# Principles for Evidence-Based and Sustainable Food System Innovations for Healthier Diets

**DOI:** 10.3390/nu14102003

**Published:** 2022-05-10

**Authors:** Chiara Ferraboschi, Jimena Monroy-Gomez, Breda Gavin-Smith, Kalpana Beesabathuni, Puja Tshering, Srujith Lingala, Neha Bainsla, Daniel Amanquah, Priyanka Kumari, Kesso Gabrielle van Zutphen, Klaus Kraemer

**Affiliations:** 1Sight and Life, P.O. Box 2116, 4002 Basel, Switzerland; chiara.ferraboschi@sightandlife.org (C.F.); jimena.monroy@sightandlife.org (J.M.-G.); breda.gavin-smith@sightandlife.org (B.G.-S.); kalpana.beesabathuni@sightandlife.org (K.B.); puja.tshering@sightandlife.org (P.T.); srujith.lingala@sightandlife.org (S.L.); neha.bainsla@sightandlife.org (N.B.); daniel.amanquah@sightandlife.org (D.A.); priyanka.kumari@sightandlife.org (P.K.); kesso.vanzutphen@sightandlife.org (K.G.v.Z.); 2Department of Human Nutrition & Health, Wageningen University & Research, 6708 PB Wageningen, The Netherlands; 3Department of International Health, Johns Hopkins University, Baltimore, MD 21218, USA

**Keywords:** food system, innovation, nutrition, principles, diet, scale-up, evidence-based, multisectoral, inclusiveness, context, accountability

## Abstract

Climate change, rapid urbanization, war, and economic recession are key drivers of the current food systems’ disruption, which has been exacerbated by the COVID pandemic. Local, regional, and global food systems are unable to provide consumers with nutritious and affordable diets. Suboptimal diets exacerbate the triple burden of malnutrition, with micronutrient deficiencies affecting more than two billion people, two billion people suffering from overweight, and more than 140 million children who are stunted. The unaffordability of nutritious diets represents an obstacle for many, especially in low- and middle-income countries where healthy diets are five times more expensive than starchy staple diets. Food system transformations are urgently required to provide consumers with more affordable and nutritious diets that are capable of meeting social and environmental challenges. In this review, we underline the critical role of innovation within the food system transformation discourse. We aim to define principles for implementing evidence-based and long-term food system innovations that are economically, socially, and environmentally sustainable and, above all, aimed at improving diets and public health. We begin by defining and describing the role of innovation in the transformation of food systems and uncover the major barriers to implementing these innovations. Lastly, we explore case studies that demonstrate successful innovations for healthier diets.

## 1. The Role of Innovation in Food System Transformations

Many factors have hindered food systems from providing healthy diets and nutritious foods. These included the rapid growth in global population, estimated to reach 9.7 billion people by 2050 [1], and climate change that exacerbates the pressure on land, water resources, and soil [2]. In 2020, COVID-19 and its consequences became an additional threat to global food systems. Food insecurity affected another 118 million people in the last year due to national lockdowns around the globe [3], further compromising the global nutritional crisis. Nutritious diets are not affordable for three billion people worldwide [4], and the triple burden of malnutrition is increasing. Stunting, wasting, and overweight affect at least one in every three children, whereas hidden hunger (micronutrient deficiencies) affects at least every second child [5].

Food system innovations (FSIs) are proposed as a game-changer to address these challenges [6]; however, a framework that leads implementors to design food system innovations for healthier diets is currently lacking. In this review, we identify six core principles needed to design an evidence-based and sustainable FSI. To do so, we provide an overview of the gaps FSI faces in order to deliver nutritious foods, we define the main challenges for implementing FSI, and we propose a new definition for FSIs for healthier diets. Finally, we present two case studies of FSI that align with the proposed principles.

## 2. What Does Food System Innovation Look Like?

### 2.1. Innovation, Food Systems and Nutrition, and Their Meaning

Innovation has become a buzzword within the food system community in the past decade. In English, innovation refers to introducing new things, ideas, or ways of doing something [7]. 

Within the food system context, innovation is considered one of the main drivers for transformation to achieve sustainable, equitable, environmentally friendly, and healthier food systems [6,8,9]; however, it remains unclear what innovation in food systems looks like.

This paper adopts the food systems’ definition proposed by the High-Level Panel of Experts (HLPE). According to the definition, the food system “gathers all the elements (environment, people, inputs, processes, infrastructures, institutions, etc.) and activities that relate to the production, processing, distribution, preparation and consumption of food and the output of these activities, including socioeconomic and environmental outcomes.” The report provides a food system framework that aims to achieve nutrition and health outcomes [10]. In the current literature, however, few highly cited food system reports and studies include nutrition (undernutrition, micronutrient deficiencies, overweight, and obesity) as the outcome. A recent review showed that the majority of highly cited international studies prioritize food production and food supply over nutrition as primary outcomes of food systems [11]. In addition, none of the major food system reports refer to FSI [12,13,14] or FSI for healthier diets [15,16,17].

To present FSIs for healthier diets as a solution to address the current nutrition challenges of food systems, it is essential to first unpack the relationship between the food system, nutrition, and innovation. Second, it is necessary to provide principles for designing an evidence-based and sustainable FSI for healthy diets and a new definition for this concept. This paper attempts to provide an in-depth analysis on the concept of FSI for diets and its applications.

### 2.2. Overview of Food System Innovations for Healthy Diets

Despite the lack of an agreed definition of FSI, the term has become widespread in the current scientific literature and has broad applications; FSI can refer to innovations in technologies (plant breeding, animal health), in policies (lowering the price of healthy foods), and in production (edible insect farming, vertical agriculture) [18]. Overall, there is agreement that FSI requires a multidisciplinary approach and the participation of different contributors, such as the private sector, the public sector, and civil society [8].

Given the limited examples of FSIs for healthier diets in the literature and their high variability [19,20], there is an urgent need to define a framework that can guide stakeholders across the food system to provide healthy diets. This will be timely to support commitments made throughout the Nutrition Year of Action (2021) [21].

There are currently no guidelines nor principles for implementors to develop FSIs for healthier diets. The available guidelines tend to address policymakers. For example, the Voluntary Guidelines on Food Systems and Nutrition of the Committee on World Food Security (CFS) primarily target and support governments to design evidence-based public policies to address malnutrition. The guidelines provide governments with a comprehensive and evidence-based reference point [22]. Whilst we acknowledge the critical role of policy in driving food systems’ transformation; we argue that a multisectoral approach that also includes practitioners is needed to achieve multiple objectives within the food system [23].

In summary, guidance should be provided to food system actors to design evidence-based and sustainable FSIs able to address all types of malnutrition. Furthermore, there is no definition of an FSI for healthier diets that reflects the needs of the implementers. In the next paragraph, we provide an overview of the challenges in implementing FSI. Finally, we propose a framework to design FSIs for healthier diets and a new definition of this concept. 

## 3. Factors to Consider While Designing a Food System Innovation

### 3.1. Complexity and Dynamism among the Food System, Nutrition and Health

First, the complex interactions between the food system, nutrition, health, and climate change make it challenging to design an FSI [24]. Nutrition and health outcomes are influenced by multiple factors, which makes it difficult to tease out and identify relevant entry points to bring a positive change or solution [25]. At the individual level, for example, the nutritional status is influenced by the consumption of single food components or other biological, socioeconomic, and psychological determinants [25]. From the environmental aspect, when developing FSIs to improve nutrition and health outcomes, it is crucial to consider solutions that will not cause environmental degradation [24]. Moreover, for the food system as a whole, a multidisciplinary team and different actors from the private, public, and non-governmental sector must work toward a coherent objective and align priorities to design an FSI [25]. This can be challenging as all stakeholders have different personal and institutional interests. Nowadays, innovation must adopt a holistic approach that will allow the solution of linear cause–effect interactions as well as more-complex and systemic interactions that consider the environment and socioeconomic context.

Second, food systems are dynamic, and nutritional and public health issues are changing at a rapid pace. Under these conditions, the rapid advancement of technology may not be sufficient to address these challenges, as providing robust evidence-based solutions from the development stage to validation and scale-up is time consuming [8,26]. For example, in the past decade, plant-based products have been proposed as a sustainable and healthy innovation compared to livestock [27,28]. Although the concept is not new and was already introduced in the 1960s [29], there are still evidence gaps around their environmental, nutritional, health impact, and safety characteristics. Another example relates to lab-grown meat that brings a potential solution to decrease livestock production. Cell-culture-related challenges around large-scale production, food safety standards, and expensive production inputs are still under discussion [30].

Lastly, complexity and dynamism within the food system can generate feedback loops and synergies, which can also influence the outcome of an FSI. Trade-offs among different components of the food system should, therefore, be considered. Indeed, trade-offs between food safety, affordability, and sustainable innovations toward food security and positive nutrition outcomes must be prioritized, taking into account the local context, especially in LMICs [10]. In addition, the environmental and nutritional impact of food remains a controversial topic within food systems [31]. To minimize trade-offs related to this topic, life cycle assessment (LCA) tools can support decision-making in food systems [32].

### 3.2. Finance and Regulation toward Entrepreneurship across the Food System

To implement an FSI, enabling environments for entrepreneurs, such as sustainable finance and regulation, are needed. From a financial aspect, the main concern is to obtain adequate and sustainable financing from the research stage, through development, to scale-up of the innovation [33]. FSIs linked to financial schemes can represent an opportunity to develop solutions to overcome financial constraints [34]. Grants, outcome-oriented finance, and productive asset finance are crucial to allow small and medium enterprises (SMEs) in the food sector to grow capacity. Recently, blended models for funding and crowdfunding have been used as an innovation for financial sustainability across SMEs in delivering nutritious foods [35,36]. At the public level, public development banks have been pointed out as key drivers in food system transformation, to channel investments in sustainable practices and innovations as well as to reshape the policy environment [37].

In terms of policy, considering the dynamic and complex nature of the food system, policymakers must react faster than ever to provide an enabling environment for FSI. Some positive changes have been made, yet meaningful transformation at the policy level to enhance the FSI is still required [38]. Policies must be designed at the upstream level based on the current evidence and learnings from previous implementation, instead of adapting existing policies that do not adopt an integrative approach [39]. For example, policymakers might consider FSI as an opportunity to improve the quality and quantity of food composition databases (FCDs), especially in LMICs, where FCD scarcity hinders the assessment of objective macro- and micronutrient intake of individuals. Filling this gap is essential to implement effective nutritional policies and programs [40]. In Ghana, a public–private partnership (PPP) launched on the market a new seal for products (porridges, cereals, biscuits) whose nutritional composition provides women a balanced micro- and macronutrient intake to fight the double burden of malnutrition. The nutritional composition of such products follows a nutrient profile scheme suitable for LMIC settings [41]. Therefore, identifying an FSI concept can also facilitate the introduction of healthier products on the market while gathering new data for FCD and supporting the introduction of new nutritional policies [42].

In summary, we highlighted key elements that should be considered when designing FSIs. Those factors were used as a starting point for the identification of the principles needed to design evidence-based and sustainable FSIs for healthy diets, described in the next paragraph.

## 4. Principles to Design Sustainable and Evidence-Based Food System Innovations for Healthier Diets

Based on the aforementioned challenges in implementing an FSI and the methodology described further below, six principles were established. A principle is a general or scientific law that explains how something works or why something happens [43]. The principles aim to demonstrate how different components of the food system interact with each other to create synergy toward evidence-based and sustainable FSI for healthier diets. The principles seek to provide a roadmap for designing FSIs for healthier diets.

We adopted a hybrid methodology to select the principles, namely a (1) brainstorming session among food systems experts and (2) a scientific review. The brainstorming session was held among eight food system and nutrition experts who have extensive experience in the field of implementation research, PPPs, and social business models to design and implement FSI. The principles were identified based on this global knowledge, expertise, and practical experience in implementation. Experiential knowledge was backed up and complemented with scientific literature in a systematic way. This methodology allowed us to identify six key principles for designing FSIs for healthier diets. The principles are mutually exclusive and are gathered under the acronym SEMICA. Figure 1 gives an overview of the identified principles.

### 4.1. Principle 1: Scaling-Up Platform

Innovations can be scaled through private sector approaches or through self-generated revenues and market-based investments. Other scaling-up platforms include the public sector through support from government, not-for-profit approaches supported by philanthropy, or hybrid approaches such as PPPs. Initial soft funding, also known as patient capital, is needed to tolerate unpredictable returns, the risks, and uncertainty involved in scaling a novel product or technology or model. There are some successful examples in the form of microfinance institutions that provide livelihood loans to smallholder farmers or grassroots food enterprises. Other examples include philanthropic or international organizations’ funding toward subsidy such as vouchers for purchasing nutritious products or government subsidy for solar technology toward food storage and machinery to increase adoption of renewable energy. Such flexible funding can dismantle barriers to scale by minimizing the high start-up costs, developing capabilities or building infrastructure that can support many enterprises at the same time. Additionally, the government and regulators have an important role to play in reforming specific policies and regulatory restrictions that make it difficult for innovations to compete with incumbents. For example, taxing unhealthy foods paves the way for more innovative foods to enter the market. Public sector actors also have a unique role in creating commonly used infrastructure. It could be in the form of physical infrastructure, such as the construction of food warehousing, cold storage, and logistics networks, or social infrastructure, such as the formation of self-help groups and cooperatives, thereby empowering them to run viable enterprises. Leveraging individual strengths is at the core of PPPs, which have a potential for an extensive transformation. In Rwanda, Africa, Improved Foods is one such example that mobilized private capital, technology, and know-how to build local production facilities. It also supports thousands of smallholder farmers while the government and the World Food Programme pooled demand and activated the distribution of nutritious foods to reach the most vulnerable groups [44] (Figure 2).

### 4.2. Principle 2: Evidence-Based Nutrition Innovation

A healthy diet promotes health and prevents disease. It should also provide adequacy without excess of nutrients and health-promoting substances from nutritious foods and avoid the consumption of health-harming substances [45]. In this light, FSIs for healthier diets need to fill the nutritional gaps of the selected beneficiaries without further compromising their nutritional status according to the no-harm principle [46]. Therefore, an evidence-based nutrition approach is needed to first identify the desired nutritional outcome of the FSI. Secondly, it helps to develop a monitoring and evaluation framework for the innovation while generating new data and strengthening scientific evidence [47], which is essential to support policymakers in adopting new policies based on the latest available scientific evidence. The Lancet framework for actions to achieve optimum fetal and child nutrition and development represents a milestone for the global nutrition community. It provides a complete, evidence-based picture of the health, nutritional, and behavioral determinants for women of reproductive age (WRA) and infants [45]. The framework can be considered as a starting point for implementors to select the most appropriate nutritional evidence-based rationale for the FSI for healthier diets. Furthermore, it facilitates the tracking of the interconnected factor related to the nutritional gap identified for the FSI. Additionally, depending on the target population, implementors can also take into consideration evidence from other scientific sources such as systematic reviews and randomized clinical trials and apply critical appraisal of the implementation considering their targeted population and the context.

Evidence-based solutions can increase the positive impact on nutrition interventions. Additionally, implementation science can shape innovations based on the local context. Tumilowicz et al. developed the Society for Implementation Science in Nutrition (SISN) framework. The methodology guides implementation planning by considering five domains (Figure 3) whose characteristics, capacities, dynamics, and adjustments can influence the quality of implementation of evidence-based nutrition interventions or innovations. The SISN framework suggests adopting the triple-A cycle proposed by UNICEF before, during, or after the implementation: assess the problem, analyze its causes, and take action based on the analysis of the different domains [48]. The principle “Evidence-based nutrition innovation” combines evidence-based innovations and implementation science to potentially strengthen food system innovations toward healthy diets.

### 4.3. Principle 3: Multisectoral Linkages

The United Nations Food Systems Summit identified multi-stakeholder platforms and engagement as a “game-changing solution” to achieving sustainable food systems [49]. The High-Level Panel on Food Systems and Nutrition identified the necessity of collaboration to create the systemic shift required to change how we produce, supply, and consume food [50]. Such linkages across the value chain can lead to alliances and coordination of initiatives and, thus, increased impact [51].

It can be argued that there is no absence of ideas. Indeed, many approaches, pathways, and initiatives have been put forward as potential solutions including, e.g., more-favorable conditions for small-scale farmers, more-sustainable agricultural practices such as agroecology, a reduction in food loss and waste, more-equitable trade policies, and regulation supporting an enabling environment to produce and consume more nutritious foods. However, for such solutions and approaches to transform how food systems function, a more concerted focus on the “how” and contextualization of solutions is required. Multi-stakeholder engagement provides this opportunity and platform. Through multi-stakeholder collaboration, all actors have an opportunity to see the whole food chain, identify their role in that chain, contribute, and catalyze collaboration and innovation. 

An example of the cross-cutting nature of nutrition is the interconnection between access to clean water sanitation and hygiene (WASH) and rates of malnutrition. Estimates suggest that 50% of malnutrition is associated with repeated diarrhea or intestinal worm infections as a result of unsafe water, poor sanitation, or insufficient hygiene [52]. Thus, nutrition interventions must incorporate WASH measures and engage with relevant actors outside of their sector to reduce malnutrition rates. Several initiatives involving the private sector have also made valuable contributions to improving nutrition outcomes through product reformulation, improved labeling standards, and restrictions on marketing and distribution to vulnerable groups. Governments should take the lead as the legislative and standard-setting body, while convening and pooling the resources, knowledge, and expertise of different stakeholders. The more diversity there is at the table, the more likely an impactful solution will emerge.

### 4.4. Principle 4: Inclusiveness

No one left behind is one of the six guiding principles of the United Nations Sustainable Development Cooperation Framework [53]. It is also the cross-cutting element of Agenda 2030 and the SDGs [54]. Leaving no one behind refers to the need to reach people who are underprivileged and discriminated, as well as individuals or families who are marginalized or excluded.

First, FSI has the potential to reduce social and economic exclusion and inequalities within the food system, ensuring that goods and services are accessible and affordable to the most vulnerable [55]. This means that applying an FSI to target the most vulnerable would reduce socioeconomic barriers while providing financing, technical, and social support to marginalized people [56] (see principle 1). Second, an FSI must be tailored to the needs of those who need it most. An inclusive FSI should be consumer centric and identify and understand the needs of the most vulnerable end consumers. From the literature, the reverse thinking approach proposes consumer insights and the food environment as a starting point for innovations, to shift consumption toward healthy diets. However, little is known about consumer-choice drivers in low-income countries [57]. FSI must generate demand for nutritious foods; it means supporting people with healthier choices and ensuring that these are valued and accessible according to their social norms, emotions, and decision-making environment [58]. For example, Nutri’zaza—a social business fighting against children’s malnutrition in Madagascar—has developed a network of baby restaurants (hotelin’jazakely) to distribute food prepared in the form of ready-to-eat fortified porridge. These baby restaurants meet the needs of low-income families in poor neighborhoods in Madagascar. Previous research shows poor complementary feeding practices and more significant difficulties of households in urban areas to prepare a meal suitable for young children than in rural areas (time, accessibility of products, and cost.) Therefore, projects aimed at offering ready-to-eat porridges help overcome some of these barriers that are linked to food preparation. Sales assistants provide to children and mothers nutritional advice and can monitor children’s weight and, when necessary, refer them to the nearest health center [59].

In summary, designing an FSI that meets the needs of the most vulnerable ensures that nutritious food is affordable and available to those who need it most. Furthermore, an inclusive FSI protects against the increment of socioeconomic inequalities among beneficiaries, in particular, consumers in LMICs who cannot afford nutritious diets [4]. Designing an FSI that embeds the principle of inclusiveness can contribute to poverty eradication, end discrimination and exclusion, as well as reduce inequalities [60].

### 4.5. Principle 5: Context Specificity

One size does not fit all. The food system in general and the FSI should be able to capture context-specific needs. The major discussions about food systems and their complexity often refer to global food systems. When consulted, local and regional food system experts provide different insights from the global outlook. For example, food availability was identified as the main challenge in food systems in East and Southern Africa, while income was considered the main one in West and Central Africa [61]. Additionally, the environmental impact of the innovation should be evaluated against context-specific challenges. For example, despite livestock production representing an environmental threat to the planet, consumption of food from animal sources is essential to address stunting in LMICs [62]. This means that the reduction or increase in ASF consumption can be the solution to FSI in different context-specific settings, according to the nutritional priorities of the target populations. Other context-specific elements that the FSI must include in this principle are culture and identity. For example, India is a diverse land with at least 15 different agroclimatic zones that range from coastal areas rich in seafood to fertile mountainous areas conducive to horticulture and staple production. In LMICs, culture plays a crucial role in determining food habits and patterns [63]. India is predominantly a staple-consuming country with only about less than 30% of the population consuming meat [64]. This statistic has seen an increase over time due to the rising urbanization, enhanced disposable incomes, and stronger market linkages.

Unfortunately, in West and Central Africa, as in many other LMICs, the burden of undernutrition is found to be disproportionately high in vulnerability pockets, particularly rural areas [65], and therefore, a more-contextualized and customized solution is required for each segment of the population.

### 4.6. Principle 6: Accountability and Transparency

Developing accountability and transparency practices can enable stakeholders in the food system to gain robust and independent insights into the success and failure of the program. However, in emerging-market contexts where nutrition programs are concentrated, there is a data vacuum. Against this backdrop, it becomes difficult to increase accountability and improve program decisions [66]. Therefore, it is imperative to assess the FSI and its impact through an unbiased third-party evaluation process, which helps to identify the strengths and weaknesses of the FSI and areas for improvement. This helps keep the innovation and all its collaterals on track, at a process level as well as an objective level, leading to better transparency, accountability, and efficient resource utilization. Furthermore, high-quality data providing the right inputs on the right indicators at the right time can galvanize decision makers. For example, Chhattisgarh is a highly populated state in India, with a history of poor development indicators. This state performed spectacularly according to data collected by a national survey in 2016. From being the state with the fourth worst proportion of stunted children, Chhattisgarh showed the highest stunting reduction in just 10 years. This stellar data point galvanized the Government of India to set similar targets for the entire country, as part of the National Nutrition Mission, which was released in 2018 [67].

### 4.7. A New Definition of Food System Innovations for Healthier Diets

A food system innovation for healthy diets is defined as a policy or regulation, an institutional process, a change in knowledge, a technology, or combination thereof that is either not used or not widely used within a food system but has the potential to change diets on a wider scale [20]. The SEMICA principles seek to guide food system actors in designing an evidence-based and sustainable FSI for healthier diets. The identification of the principles serves as a starting point for an agreed definition among the food system community and can complement the above definition by providing a practical approach and solution to FSI to address the various challenges of our food system. We, therefore, propose an updated definition for this concept and define an FSI for healthier diets as an intervention that meets the SEMICA principles.

## 5. Case Studies

The following paragraph presents two case studies of FSIs for healthier diets that meet the SEMICA principles.

### 5.1. SAL Case Study: Egg Hub Malawi

Egg Hub Malawi is an FSI that meets the SEMICA principles. Eggs are cheap, available, and frequently consumed by young children in high- and middle-income countries; they are expensive, scarce, and rarely consumed by children in much of Africa and South Asia. Trying to improve the productivity of individual small-scale producers is unlikely to significantly improve egg consumption at the national level. An egg hub model that aggregates clusters of rural producers might increase access to eggs for many poorer countries [68]. By aggregating farmers into small groups, recurring capital and program costs for each egg produced are significantly lower than the inefficient, micro-scale backyard production systems [69]. The Egg Hub is a multipronged approach that is self-sustaining and replicable under local conditions and helps to achieve household nutrition security in low-resource rural and peri-urban settings. An integrated marketing campaign supports increased production, driving strong behavioral change among communities and increasing egg consumption. Thus, it holistically impacts the value chain by creating a middle-market, combining the economic efficiencies and best practices of commercial farming but benefiting the vulnerable by organizing and building capacity at the grassroot and smallholder levels and a nutrition intervention targeting women and men. The project aims to improve the availability, affordability, desirability, and quality of eggs (Figure 4).

#### 5.1.1. Scaling-Up Methods

The hub consists of a centralized unit operated by an entrepreneur who organizes smallholder farmers into groups, provides an input package, training, and market support to sell eggs. Each unit caters to farms up to a 100 km radius, beyond which eggs are likely to break or get spoilt during transportation [70].

#### 5.1.2. Evidence-Based Nutrition Innovation

In Malawi, 37% of children under five years of age are stunted [71]. Consumption of eggs is related to a higher consumption of several key nutrients important for growth and development and better linear growth [72,73,74].

#### 5.1.3. Multisectoral Linkages

Egg Hub Malawi strengthens the local value chains of eggs from farmers to consumers. Smallholder and backyard poultry farmers face multiple challenges in Malawi. First, predominant backyard farming is riddled with high bird mortality and low productivity. Second, farm inputs are inaccessible; birds receive extremely poor nutrition and no vaccines. Third, eggs are rarely sold in markets at fair prices due to inconsistent production and no coordination among farmers. Lastly, smallholders cannot compete with commercial-scale farms on price, quality, or efficiency. To mitigate many of these challenges, Sight and Life, Lenziemill (feed miller), and Maeve (a grassroots NGO) formed a partnership to support farmers through input packages, credit, training, and access to markets. Backyard farmers were organized into small groups of five and supported to set up and develop an enterprise with a 3-year break-even period. Farmers receive a startup input package on credit, which they pay back over time and are eventually self-sustainable in 3 years. Farmer groups are encouraged to buy improved feed at wholesale rates and sell the eggs primarily in their communities. The trucks that deliver feed buy poultry input materials, i.e., maize and soya, and any excess eggs from farmers, thereby creating a holistic cycle. Farms start with 720 birds and scale-up.

#### 5.1.4. Inclusiveness

The hub in rural Malawi, where every third child or woman is malnourished [71], supports 35 farms and 175 farmers, including convent nuns, all-women groups, schools, and the urban poor living in slums [75].

#### 5.1.5. Context Specificity

Eggs continue to be scarce and expensive in Malawi—the average annual consumption per capita is only 27 eggs, compared to 180 globally [76]. The social marketing approach to increase egg consumption was carried out in eight villages. The integrated marketing campaign spread awareness about the benefits of egg consumption among mothers and pregnant and lactating women. The campaign’s tagline is More Eggs, More Smiles, since good health is seen to be synonymous with happiness [75]. The preliminary findings show that household weekly egg purchase has increased from 7 eggs to 10 eggs, and children are consuming 16% more compared to the baseline. This rise in consumption is largely driven by households who did not consume eggs at all.

#### 5.1.6. Accountability and Transparency

In 2019, the impact evaluation showed that the model led to two times more eggs available at 20% lower prices and three times higher income for the farmers [75]. In 2021, the estimated preliminary numbers suggest that ten million fresh eggs are being produced every year and that the egg hub increased the daily income of farmers by threefold and lowered egg prices by 20% for 210,000 rural poor [75].

### 5.2. Fruits and Vegetables for Vietnam (FVN) Project

The FVN project represents is another FSI that meets all SEMICA principles. The project aims to increase the consumption of fruit among low-income consumers in peri-urban/urban areas in Vietnam, Hanoi. Food system innovations improve the diversification of retail outlets, improve affordability through a client-specific coupon system, and increase acceptability of fruits through promotional campaigns involving public and private stakeholders and civil society organizations [77]. The FSI consists of the distribution of vouchers designed to purchase specific kinds of fruits at the local market for a period of five months. The coupons were delivered once a week and were valid for a wide range of vendors at the local market [77,78,79].

#### 5.2.1. Scaling-Up Methods

Thirty-five fruit and vegetable sellers were affiliated with the voucher system. They received training on nutrition consultancy for customers and more-appealing displays.

#### 5.2.2. Evidence-Based Nutrition Innovation

A recent study showed that Vietnamese people only consume between 66% and 77% of the recommended daily intake of fruits and vegetables [79]. WHO recommends the consumption of 400 g per day per person to promote a healthy diet and prevent non-communicable diseases and micronutrient deficiencies [80].

#### 5.2.3. Multisectoral Linkages

The starting point of the intervention is consumers, followed by retailers and farmers [77]. The FVN project aims to assist small businesses from lower socioeconomic classes in selling safe vegetables in urban areas. This initiative resulted from a series of co-creation workshops with the participation of consumers and small retailers in the project area. Part of the project consists of strengthening the value chain of selected vegetables and guaranteeing consumers safe and affordable vegetables. The sellers were involved in the collection of vegetables at a location to decrease the transportation costs and they were provided with equipment such as safe veggies signboards, shelves, and lanyards along with support for implementing discount cards for selling safe vegetables to people with a low income [78].

#### 5.2.4. Inclusiveness

In total, 400 low-income households from the peri-urban area of Hanoi were selected as beneficiaries of the innovation [80]. Every low-income household of the project receives a weekly coupon with a value of VND 30,000 or VND 60,000, valid for two weeks to purchase vegetables.

#### 5.2.5. Context Specificity

A context-specific analysis showed household preferences and habits were key determinants to support the consumption of fruits and vegetables [81]. In Hanoi, 80% of families do not meet the recommended daily consumption of fruits and vegetables recommended by the WHO [79]. Identified determinants were preferences and health knowledge [81].

#### 5.2.6. Accountability and Transparency

Twelve weeks after the voucher’s implementation, the voucher redemptions reached 80% of the households [79]. Since December 2020, 300 consumers have joined Rikolto’s loyalty card program, 60–70% of them have increased from one to two turns of purchase per week [82].

## 6. Conclusions

At present, stakeholders in the food system have no clear guidance on how to design evidence-based and sustainable food system innovations that are economically, socially, and environmentally sustainable and provide healthier diets. To address this gap, this paper introduced six principles (SEMICA) aimed at supporting these actors in doing so. It also proposes an adapted definition of food system innovation for healthier diets. We argue that clear guidance on the design of FSIs can complement academic and political discussions around global food systems and facilitate the achievement of healthier diets for the global population, especially in LMICs. Furthermore, we believe that this paper can stimulate a discussion around the meaning and importance of FSI for healthier diets and its applications and also provide a better understanding of this concept within the food system community.

## Figures and Tables

**Figure 1 nutrients-14-02003-f001:**
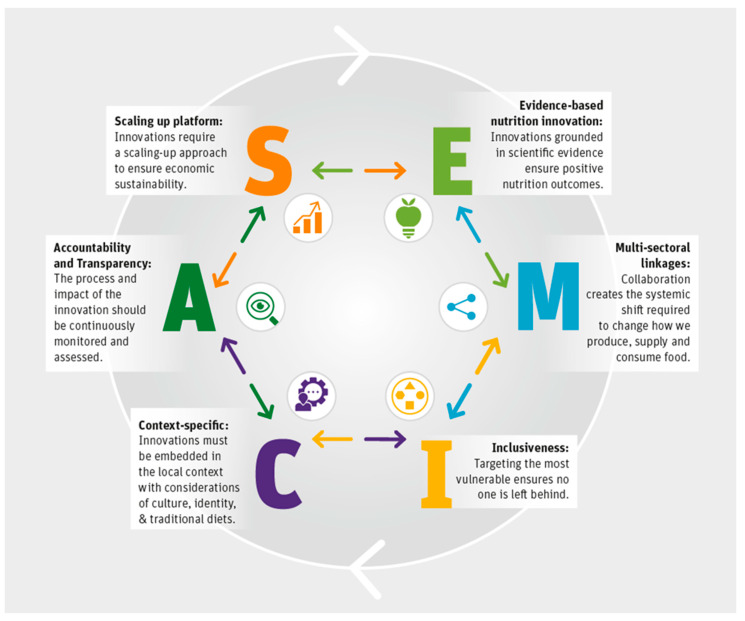
SEMICA principles for designing evidence-based food system innovations for healthier diets.

**Figure 2 nutrients-14-02003-f002:**
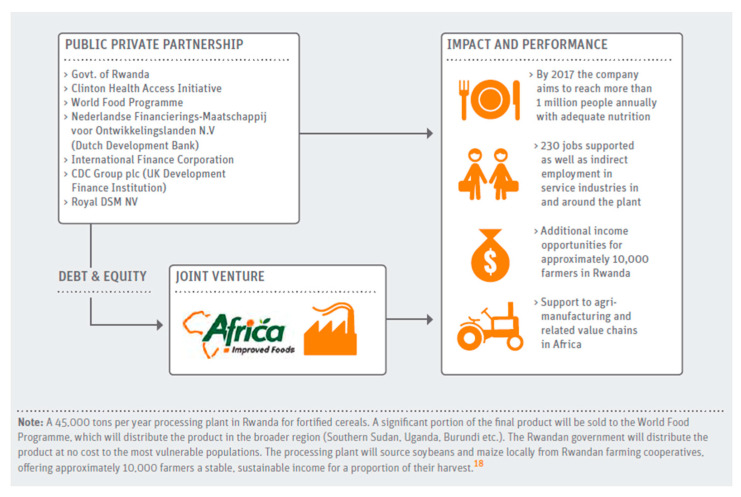
Impact of investing through a public–private partnership model in Rwanda [44].

**Figure 3 nutrients-14-02003-f003:**
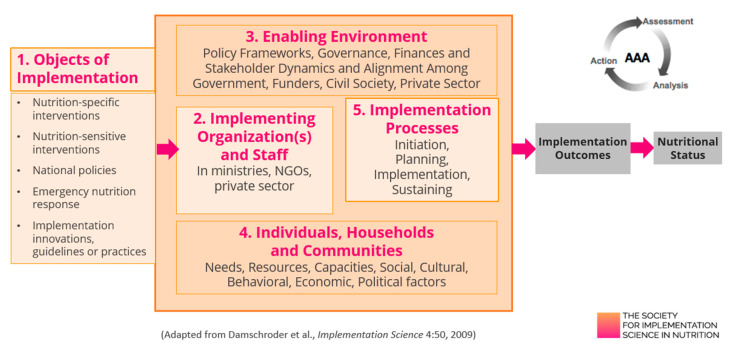
Domains of implementation science in nutrition framework. Source: SISN website—https://www.implementnutrition.org/sisn-framework/ (accessed on 12 December 2021).

**Figure 4 nutrients-14-02003-f004:**
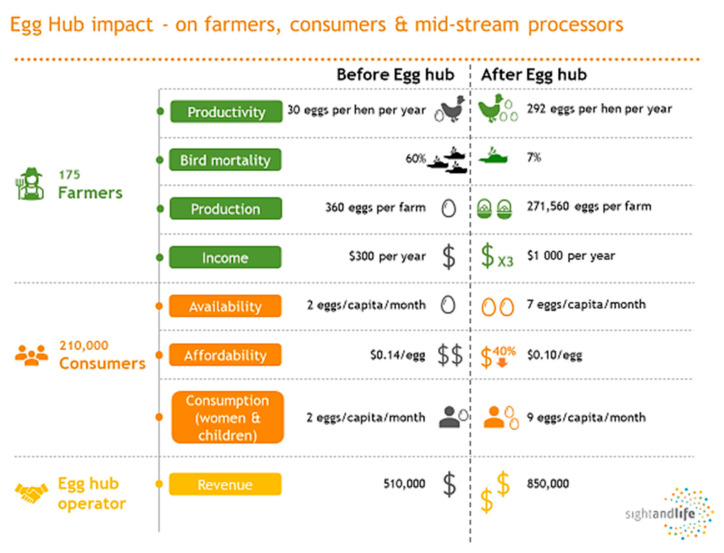
Malawi egg hub impact in 2021.

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
