# Peer review of "Principles for Evidence-Based and Sustainable Food System Innovations for Healthier Diets"

_nutrients, 2022, doi:10.3390/nu14102003_

Round 1

Reviewer 1 Report

The subject is extremely important. The paper should be published. It is written in very good English and the presentation of the subject is very clear. The literature used in this paper is sufficient. More details should be presented how the Egg hub Malawi was implemented. Hub project was launched in 12 villages in central Malawi to set up and develop bird poultry farms with 3-year break even period. This project has brought together various partners to support poultry farmers, who are organized into groups of five. In addition to receiving specialized feed, the groups also receive all important vaccinations for the birds, which are ready to lay when they arrive at the farms, training, and continuous supervision.

More details should maybe be presented also how the Fruits and Vegetables for Vietnam and Nigeria (FVN) project was performed.

Author Response

Dear reviewer,
Thank you for giving us the opportunity to submit a revised draft of the manuscript “Principles for evidence-based and sustainable food systems innovations for healthier diets” for publication in the Journal of Nutrients. We appreciate the time and effort that you and the reviewers dedicated to providing feedback on our manuscript and are grateful for the insightful comments on and valuable improvements to our paper. We have incorporated all the suggestions. Those changes are highlighted within the manuscript.

  • Please see in section 5 of the paper, the extra information requested about each case study.  

Reviewer 2 Report

The context should be better introduced. The authors should mark the importance to have food composition data for complex  food matricies, food preparations and recipes and related references should be added such as:

Durazzo,et al (2017). Nutritional composition and antioxidant properties of traditional Italian dishes. Food chemistry218, 70–77. 

The authors should mark the novelty character of this paper.

The description and discussion of case studies should be implemented.

Author Response

Dear reviewer,
Thank you for giving us the opportunity to submit a revised draft of the manuscript “Principles for evidence-based and sustainable food systems innovations for healthier diets” for publication in the Journal of Nutrients. We appreciate the time and effort that you and the reviewers dedicated to providing feedback on our manuscript and are grateful for the insightful comments on and valuable improvements to our paper. We have incorporated some of the suggestions. Those changes are highlighted within the manuscript.

  • Please see in section 3.2 of the paper, the extra information requested about food composition data. Indeed, there is a connection between FSI and food composition data. We included further literature based on studies from Low and Middle-Income countries, in line with the title of the Special Issue “Sustainable Food Systems for Nutrition in Low Resource Settings”  of our article and with Sight and Life focus.   
  • Please see in section 5 of the paper, the extra information requested about each case study.